# Cold Roll Forming Process Design for Complex Stainless-Steel Section Based on COPRA and Orthogonal Experiment

**DOI:** 10.3390/ma15228023

**Published:** 2022-11-14

**Authors:** Jing Wang, Hua-Min Liu, She-Fei Li, Wan-Jun Chen

**Affiliations:** 1College of Materials Science and Engineering, Jilin University, Changchun 130025, China; 2Roll Forging Research Institute, Jilin University, Changchun 130022, China; 3Battery System, Nio Co., Ltd., Shanghai 201805, China

**Keywords:** roll forming, finite element analysis, stainless steel, parameter optimization

## Abstract

Cold roll forming can fabricate products with complex profiles, and its parameter optimization can achieve high quality and improved precision of products. In this paper, taking the side shield as a typical product, the cold roll forming of a complex section of stainless steel SUS301L-ST is analyzed, establishing a 3D finite element model by using the professional roll forming software COPRA. We propose a floating roll device for complex sections with asymmetry and large depth. We use an orthogonal experiment to obtain the inter-distance between rolls, friction coefficients, the diameter increments, and line velocities to investigate the effects on the maximum longitudinal strain of the edge. Results show that the diameter increment has the greatest influence on the maximum strain, and its increases can reduce the strain. The inter-distance value needs a suitable range. A small value is not conducive to the release of elastic deformation, while a large value will cause unexpected displacement and increase the cost. The friction coefficient increases; although it helps to reduce the strain, it will cause scratches and other defects on the stainless steel. The increase in velocity increases the strain. We derive the optimal parameters for the complex section, providing a theoretical basis for practical production.

## 1. Introduction

Cold roll forming is a processing technology used to manufacture metal products, which can produce high-quality products, shorten the development cycle, improve efficiency, and save materials and costs, so it has a number of applications, such as automotives, railroads, aviation, and ventilation [1]. The raw materials are carbon steel, stainless steel, aluminum alloy, etc., among which stainless steel is the major material of railways; compared with others, it has the advantages of high strength and stiffness, high corrosion resistance, and is economical and durable. For stainless-steel materials, the surface quality is an important evaluation criterion [2]. Zhang researched the surface defects of automotive decorative bars with stainless steel, and solved the problem of fine ripples by modifying the forming angle, line velocity, and friction coefficient [3].

Cold roll forming belongs to large elastoplastic deformation, which is not only transverse and longitudinal deformation, but is also influenced by external friction, hole geometry constraints, the shape and size, and the material’s hardening [4]. However, industrial practice relies on experience, and there are no accepted rules that could cover various profiles. Correction of defects such as edge waves and wrinkles is performed by trial-and-error to bring the product into line with the dimensional requirements, which consumes materials and time [5]. With increased attention, researchers have increasingly developed numerical simulations, especially regarding the parameters. Numerical simulation of cold forming shows that the deformation changes from elastic to plastic deformation and the bending stress increases and then decreases [6]. Setting up symmetrical simulations for cross-sections such as corrugated plates greatly reduces the time and costs [7]. Amin Poursafar and his team built a simulation program to examine the spring-back of the roll forming process. They concluded that the thickness increase causes the forming length to decrease and the plastic anisotropy affects the spring-back [8]. Longitudinal strain is the criterion for predicting defects in the edges [9]. The greater the flange width, forming angle, and yield strength, the greater the edge of longitudinal strain. The greater the thickness and the inter-distance value, the smaller the strain. The web width has almost no effect [9,10,11]. The friction coefficient has less influence on the forming angle and affects the surface quality [12]. The angular order of design has an effect on the forming accuracy. The forming quality is greater for the constant arc length forming method in comparison to the constant radius method [13]. Optimized parameters can reduce strains, thus reducing the defects and increasing productivity [14]. Ce Liang discussed the effect of the thickness of the sheet, the height of the flange, and the forming speed on the edge wave, noting the maximum effect of thickness and minimum effect of speed [15].

In addition, the parameters can affect the spring-back, which is an important factor affecting whether the product meets the precision requirement. In the shape and dimensions of the cross-section, spring-back increases with increasing flange width and edge height, and the distance decreases with increasing strip thickness and web width [16]. Zheng concludes that the greater the thickness, the greater the increase in stress and strain and the smaller the spring-back [17]. In Q.V. Bui et al.’s work, they concluded that the spring-back increased with the yield strength, and the effect of the yield strength was greater in the thinner material [18]. Cheng Jiao-Jiao et al. used COPRA to simulate two types of flower for a high-strength steel cap-shaped section, compared them, and proposed a new UDT-type angle change method, achieving high-precision spring-back control through finite element simulation [19]. Jia Weitao et al. optimized the parameters of the stainless steel in a heat exchanger plate and designed the rolls according to the spring-back, allowing the product to meet the accuracy requirement [20]. Cao et al. used COPRA to simulate a round tube into an “8” shape, and explored the effect of parameters on the stress and optimized the shaped tube through adjusting the rolls [21]. Different tooling concepts are used to improve the strain distribution and hence the part quality, which can be used to form a complex shape with the minimum tooling requirements [22]. For accuracy between simulation and experiments, Kwun Sing Tsang performed measurements on the basis of surface geometry. Good agreement was observed between the simulation models and experiments, indicating the reliable application of the FE methodology for the roll forming process [23]. Cheng-Fang Liu developed a new mathematical model to analyze roll forming and verified its reliability [24]. Numerical simulation significantly reduces input costs and increases productivity in cold roll forming. Li et al. performed a numerical simulation for 301L stainless steel in cold roll forming. They concluded that the effects of front and back tension on the strip flatness are the most obvious compared with the friction coefficient and line velocity [25].

The above literature shows the influence of different kinds of parameters on simple and symmetrical sections. Meanwhile, the effects of these parameters on complex and asymmetrical sections are almost never predicted. This paper uses COPRA to simulate cold roll forming for sections with high asymmetry and a large depth, and to analyze the stress and strain. We propose a floating roll device, which solves the problem of oversized rolls due to excessive depth. This paper uses the orthogonal experiment to explore the influence of the inter-distance value, friction coefficient, line velocity, and roll diameter increment, and uses the statistical method to obtain the best combination of parameters. Finally, we obtain a the section with good stability and product quality requirements, and a low cost.

## 2. Materials and Methods

### 2.1. Description of Cold Roll Forming Process

Cold roll forming comprises multiple rolls with a specific profile. The metal strip moves along the longitudinal parallel rolls at room temperature, producing plastic deformation, becoming the required product. It divides the deformation process into loaded bending and unloaded spring-back. In the traditional design, CAD is used to draw, which results in a huge calculation workload and reduces the accuracy. This paper uses the integrated software COPRA, developed by Data M, to carry out the design and analysis, which can improve the efficiency and accuracy because of its ability to offer quick solutions, estimate costs, and simulate the forming process.

The roll in this paper consists of 24 stations and the assembly diagram is shown in Figure 1. When the metal strip enters the roll, the center part flows in a straight path, while the edge rises vertically and flows horizontally to the center. Due to the vertical and horizontal movement, the edges elongate, where L1 is significantly longer than L under the effect of longitudinal stretching (Figure 2). Because of this deformation, it produces various redundant variants in the metal; stretching and compressing can form defects. In addition, there is spring-back between the two stations, mainly due to the elastic deformation energy stored, which is released during unloading. Several factors influence spring-back; the high yield and tensile strength and the work hardening of the stainless steel will also exacerbate the spring-back. The larger bending radius of the profile also increases the spring-back.

### 2.2. Material Properties and Meshing

This paper studies the stainless steel SUS301L-ST formed by continuous rolls, which is widely used in railways due to its good mechanical properties and ease of manufacturing. The properties and the chemical composition of SUS301L-ST are shown in Table 1 and Table 2, respectively. The calculated width is 780.2 mm, with a thickness of 2 mm. The product was modelled using continuous forming, but, due to limited computer calculations, we set the strip length at 2500 mm, more than twice the inter-distance value, to ensure that two rolls were involved in forming at the same time. To facilitate strip entry, the model incorporates support rolls.

Using COPRA-RF, we imported the metal sheet; the front of the sheet was chamfered on both sides to allow better access to the rolls. COPRA-FEA automatically divides the mesh at the corner, which is denser compared to the straight line, and the meshes are deformed because of the chamfers (Figure 3). The contact body defines the roll as a rigid body and the sheet as a deformed body.

## 3. Flower and Roll Design

### 3.1. Flower Design

The purpose of cold roll forming is to form the required product with a minimal number of steps. Forming too quickly will produce unbearable stress and distortion, while slow forming will lead to a higher cost. Therefore, the parameters should be based on a variety of factors, such as section complexity and depth, material properties, bending angle order, surface, thickness, etc.

Because of the complex shape, as well as the high yield and tensile strength of the material, the number of stations increases, which makes forming more difficult. If it follows the traditional placement (Figure 4a), this will lead to several problems. The difference in angle between the two sides is so great that it will twist toward the larger side, and this makes the diameter increase. The roll diameter difference is too large, resulting in an enormous difference in velocity, increasing the stainless-steel surface’s scratches. We develop the placement with the sectional dimensions (Figure 4b). This arrangement places the datum line at the lowest point of the arc, and the basic centerline at the center of the arc, which can reduce the angle difference. In addition, as the diameter increases the distance, designing the side roll can effectively reduce the diameter, save the roll, and reduce the cost, making it easy to install and debug.

Using an inside-out forming sequence, we keep the forming horizontal in 1–15 stations to reduce the distance and to ensure the quality, and we form the blind angle on the right side first (Figure 5a). The arc with a radius of 350 mm is over-bent to bring the blind angle into contact with the bottom roll. The left-side arc from the first station over-bends by 10 degrees in total (Figure 5b). We show the pattern according to the above requirements in Figure 6.

### 3.2. Rolls Design

The roll design process must consider a number of defects, such as difficulties in biting, lateral movement of the strip, roll processing requirements, etc. The strip is deformed at room temperature and low speed, so the thermal expansion is ignored. The material of the roll is cold tool steel Cr12, which has high strength and resistance and is suitable for stainless steel. The chemical composition of Cr12 is shown in Table 3.

It forced the upper axis to rise because the section depth increased, making the roll volume and weight increase, leading to difficulties in installation and commission, poor stability, and other problems. We develop a floating roll device that addresses this problem. As the forming process is driven by the bottom roll, the actual upper axis can be moved up by 300 mm, which will be called the axis of the floating roll to avoid collision with the section, using bearings and sidearms to connect the floating roll to the top roll. This solves the problem of weight and volume, saving the materials, reducing costs, and increasing forming stability.

## 4. Results and Discussion

### 4.1. DTM and Analysis of the Simulation Results

After defining the roll pattern, we use the deformation technology module (DTM) to deform the initial pattern. The roll forming process is always accompanied by longitudinal strain on the edges of the sheet, which is because the edges have to cover a longer distance compared to the undeformed area. Data M introduces a linear method, including the forming length. This module is based on finite element analysis and extensive experiments to check the quality according to the geometric deformation and material properties. It solves the potential problems during the design and reduced the commissioning with DTM, which not only speeds up the trial period of new products but also reduces the material and costs.

The peak magnitude of the longitudinal strain is an important indicator of the quality. We show the strain variation curve with stations for the DTM in Figure 6. The longitudinal strain changes slightly when forming horizontally. When the depth increases in the 15 stations, the strain increases beyond the tensile strength and may lead to defects. Results were consistent with the edge wave generation theory; side waves are forced to move in both vertical and horizontal directions to produce longitudinal strain.

Figure 7 shows the equivalent stress distribution of the final product. The equivalent Von Mises stress complies with Formula (1):(1)$$\bar{\sigma }=\frac{1}{\sqrt{2}}\sqrt{{\left(\sigma_{x}-\sigma_{y}\right)}^{2}+{\left(\sigma_{y}-\sigma_{z}\right)}^{2}+{\left(\sigma_{z}-\sigma_{x}\right)}^{2}+6\left(\tau_{xy}^{2}+\tau_{yz}^{2}+\tau_{zx}^{2}\right)}=\sigma_{s},$$
where $\bar{\sigma }$ is the equivalent stress; $\sigma_{x}$, $\sigma_{y}$, and $\sigma_{z}$ are normal stress components at the *x*, *y*, and *z* plane, respectively; $\tau_{xy}$, $\tau_{yz}$, and $\tau_{zx}$ are shear stress components acting in different planes, respectively, and $\sigma_{s}$ is the yield stress.

Plastic deformation occurs in the force body when the equivalent stress reaches a certain value under deformation conditions. The stresses distribute in the contact between the roll and the strip. Stress is higher near the bend and decreases around it. The bend near the edge is more difficult to form due to the large depth and the more complex forces, which make forming unstable. Therefore, we analyze the section with the parameters of 800 mm inter-distance, a friction coefficient of 0.1, a velocity of 10 rpm, and a diameter increment of 0.6 mm as an example. The left moves inwards, due to the complex deformation forces during the interaction of the rolls. Therefore, to ensure the accuracy, the stress and strain with stations at the edge of 60 mm away from the chamfer at the top, middle, and bottom of the left and the displacement X are measured (Figure 8). When the depth increases, the stress and strain relationships are both top > middle > bottom. The volatility and peaks in the stress and strain curves at the top nodes are greater than those at the middle and bottom nodes, showing that the sides are most susceptible to side-wave defects.

Figure 9 shows a plot of the stress and displacement X with stations at the bend near the left edge. It is concentrated on the stress near the corner, with the corner being the largest and decreasing to the sides. No plastic deformation of this corner occurred during horizontal forming. When the depth increases in the 15 stations from the flower pattern, the stress increases at the bending. According to the value of the bending design, the level of stress increase is different. The gradual accumulation of stress in the subsequent stations exceeds the yield strength and plastic deformation occurs. Regular lateral displacement of the sheet occurs as the edge rises. Little lateral movement of the sheet occurs during horizontal forming and the lateral movement of the sides continues to rise as the depth increases. The reason is that with increasing depth, the sides of the sheet are constrained mainly by the side rollers, which reduces the forces applied and renders the forming more unstable.

### 4.2. Inter-Distance between Roll Stations

The increased depth of the section increases the forming length and therefore requires a greater distance between stations. In order to explore the effect of different inter-distance values on the forming, we set three cases as shown in Table 4. The sheet requires a certain distance to relieve internal residual stresses as it rises gradually. Other parameters include a friction coefficient of 0.1, a velocity of 10 rpm, and a diameter difference of 0.6 mm. Figure 10 shows the stress distribution of the different cases. The stresses are small in all three results. The front of the sheet with a small inter-distance is strained in contact with the rolls, resulting in breakage. The longitudinal strain with stations is shown in Figure 11. The figure shows that the longitudinal strain follows the same trend, with a significant increase when the edge is raised. Moreover, an increase in the inter-distance reduces the longitudinal strain. The reason is that as the edge rises and the depth increases, the edge undergoes greater displacement, requiring a large spacing to release elastic stresses. Overwise, the sheet will not be fully formed under the shape of the roll gap, which will reduce the forming accuracy. However, when the inter-distance changes from 600 mm to 1000 mm in the 15 stations, the strain increases to 0.08% approximately, because the rolls vary too much. A sudden increase in strain occurs when the spacing is changed from small to large. The sheet is less constrained by the two rolls, which causes unnecessary displacement and a loss of stability. Therefore, for complex sections, the inter-distance value needs to be changed within a suitable range in order to reduce the probability of defects.

We selected the node at the bend after unloading, and the variation in the displacement in the x-direction is shown in Figure 12a. The small arc length represents the front of the strip. The value of X displacement at the front of the sheet is larger than the end in Figure 12a. The reason is the large deformation that occurs when the front of the sheet touches the rolls. In practice, the front of the strip is usually cut off due to continuous forming. The fluctuations in the lateral deflection flatten out slightly as the distance increases, because the increased distance allows for the more adequate release of the elastic stresses, making the forming more accurate. The results of the cross-sectional comparison between the unloaded sections and the target profile are shown in Figure 12b. The differences are mainly on the deeper left side. The increased distance results in a slight reduction in the forming accuracy of the enlargement. The reason is the increased lateral movement because of the excessive distance. Therefore, the distance needs to be combined with the actual conditions and has a certain range, while considering the control of costs.

### 4.3. Friction Coefficients

In order to investigate the influence of friction coefficients on the roll forming, we set the coefficients at 0.05, 0.1, and 0.15, respectively. Other parameters include an inter-distance of 800 mm, a velocity of 10 rpm, and a roll diameter difference of 0.6 mm. Figure 13 shows the equivalent stress distribution of the different friction levels. As the friction coefficient increases, the stress distribution after unloading is similar. Figure 14 shows the longitudinal strain at the same location on the edge. Maximum longitudinal strain values and fluctuations occur at a friction factor of 0.1. The longitudinal strain fluctuates slightly when the edge is horizontal. As the coefficient increases, there are abrupt changes in strain values at the edges as the depth increases. The influence of the friction on the edge strain varies at different forming stages. Overall, the variation is minor, indicating that friction has little effect on longitudinal strain at the edges.

Figure 15a shows the transverse displacement for friction coefficients. The curve shows that an increase in the coefficient has a slight effect on the torsion. Figure 15b shows a cross-sectional comparison with the target. The difference in the forming angle on the right is small and the friction coefficient has less influence on the forming angle. However, for stainless-steel products, the most stringent requirement is the surface quality. The higher the coefficient, the more likely the roll will stick and the surface will be prone to defects, such as scratches. Thus, the influence of the coefficient is mainly in the surface quality. Processes such as laminating rolls, plating chrome, and in-line lubrication are derived for stainless steel.

### 4.4. The Diameter Increments of Rolls

The forward velocity of the sheet depends on the driving rolls. The rolls drive the sheet forward, resulting in a gradual increase in speed. We set incremental values of 0 mm, 0.6 mm, and 1.2 mm for the bottom roll diameter to stabilize the form and maintain a certain tension. Since the lower roll is driven, the top roll only needs to meet the strength requirement. Other parameters include inter-distance 800 mm, velocity 10 rpm, and friction 0.1. Figure 16 shows the equivalent stress of different diameters. The increasing diameter effectively reduces the stress and increases the forming stability. Figure 17 shows the longitudinal strain at the same location on the edge. The longitudinal strain is the greatest when the edge starts to rise and gradually stabilizes as the forming proceeds. Longitudinal strain varies considerably when the diameter is the same, and it is prone to defects. It minimizes the strain when the diameter increment increases, resulting in stable forming. Therefore, a suitable diameter increment increases the forming stability and reduces longitudinal strain.

Figure 18a shows the transverse displacement curves. The graph shows that the lateral movement is smoothest at the maximum diameter increment. Figure 18b shows a cross-section compared with the target. The increase in diameter increment results in higher forming accuracy at bends. In summary, the appropriate diameter increment means that the sheet more smoothly and effectively avoids warping or piling up because of stacking. The principle of selecting the increment is that there is no plastic deformation in the longitudinal direction, while the cost needs to be considered. For example, if the roll diameter is too large, the weight increases, and the process cost increases.

### 4.5. Line Velocity

In order to investigate the influence on line velocity, we set the velocities at 5, 10, and 15 rpm, respectively. Other parameters include inter-distance 800 mm, friction 0.1, and diameter increment 0.6 mm. Figure 19 shows the equivalent stress of the different velocities. The difference in equivalent stress is small and has less impact on the stability, because the absolute values of stresses are distributed in the range of 38.13 to 467.5 MPa. Positive and negative numbers indicate only stretching or compression behavior. Figure 20 shows the longitudinal strain at the same location on the edge. The effect of velocity on longitudinal strain is less when forming horizontally. There is a significant increase in longitudinal strain at elevated sides. When the speed is too fast, the strain rate increases, which leads to work hardening, and the internal strengthening requires greater forces during deformation.

Figure 21a shows the transverse displacement curves. The graph shows that the lower the velocity, the lower the lateral movement. Figure 21b shows a cross-sectional view. The enlarged view of the corner shows that the lateral movement has caused the material to move to the web when the velocity is maximum. The reason is that high speeds lead to insufficient restraint and elastic stress relief, resulting in large forming dimensional errors. Several factors influence the forming velocity—mainly the material properties, thickness, shape, number of stations, forming length, and inter-distance. Therefore, a suitable forming velocity requires these factors. In actual production, numerical simulation and practice are effectively combined to achieve the best quality based on a stable form.

### 4.6. Analysis of Orthogonal Experiment Results

We use the L_9_(3^4^) orthogonal array for four factors with three levels, comprising a set of nine experiments. The parameters and the levels studied are shown in Table 5. The maximum longitudinal strain at the edge of each group is calculated. The purpose is to minimize the strain. The signal-to-noise ratio (S/N) can measure the influence of each factor and level on the forming quality. The higher the S/N ratio, the smaller the strain, and the better the forming quality. The L_9_ orthogonal array, the maximum strain, and the S/N ratio are presented in Table 6. Then, the experiments used the mean value analysis. We present the range analysis in Table 4, with the range R of each parameter at different levels (Table 7). If R is large, it means that the parameter has a large impact on the indicator, and vice versa.
(2)$${\left(S/N\right)}_{i}=-10\times \log_{10}\left(\frac{1}{n}\sum_{i=1}^{n}y_{i}^{2}\right),$$
where $y_{i}$ is the calculated value for each experiment, *n* is the number of repeated experiments, and ${\left(S/N\right)}_{i}$ is the signal-to-noise ratio.

The average S/N ratio was obtained based on the S/N of the maximum strain, as shown in Figure 22a. The dashed line shows the overall mean. The curve of the R is shown in Figure 22b. The friction coefficient and the roll diameter increment on the maximum longitudinal strain are positively correlated, and the effect of line velocity is negatively related. The positive correlation coefficients are 0.99574 and 0.99564, respectively, and the negative correlation coefficient is 0.92882 through calculation, which shows that the data have a highly linear correlation for the sectional shape in this study. Inter-distance is not linear to the strain. It follows that the depth increases, and the continuous increase in the distance negatively affects the strain. Friction is usually accompanied by energy consumption, which tends to cause a temperature increase, residual stress, and surface damage at the contact body. In practice, both the roll and the workpiece surface are treated with a lubricant to remove the heat generated, while washing away the surface particles and debris to reduce scratch defects. Therefore, although the decrease in the friction increases the strain slightly, the friction should consider the surface quality of stainless steel. The increase in the diameter increment effectively reduces the strain. Friction and other factors affect the speed difference between the roll and the sheet. As shown in Figure 22b, the diameter increment has the greatest impact on the maximum strain, and the other parameters are below the average compared with it. The smaller the strain is, the better the forming quality is—that is, the larger the average S/N ratio is, the better. The maximum average S/N ratio in Figure 22a is the best of the parameters. Therefore, the optimal combination of parameters can be determined from the comprehensive analysis illustrated in Figure 22. The inter-distance is 800 mm, the friction coefficient is 0.05, the roll diameter increment is 1.2 mm, and the line velocity is 5 rpm. Based on the numerical study, the equipment of the cold roll forming production line is as shown in Figure 23a, and the actual production of sections that meet the precision and surface quality requirements is shown in Figure 23b,c, respectively.

## 5. Conclusions

This paper uses the professional cold roll forming software COPRA to design and simulate patterns and rolls. We optimized the traditional placement of the datum and basic centerline, solved the problem of uneven bending angles leading to torsion, and invented a floating roll device that effectively reduces the distance and the oversized top rolls, greatly reducing assembly difficulties and saving materials and costs. The forming method in this paper provides a reference for the design of profiles with large depths and bends, blind corners, and asymmetry. The exploration of cold roll forming parameters for complex sections allows the application of the current method to more complex profiles.

We designed L9(3^4^) orthogonal experiments by using the inter-distance, friction coefficient, roll diameter increment, and forming velocity as parameters, exploring the effect on the maximum longitudinal strain of edges using the mean value analysis and signal-to-noise ratio. The roll diameter increment has a greater influence on the longitudinal strain among these, because it effectively promotes the advancement of the profile, providing tension and making the forming more fluid, while avoiding the phenomenon of stockpiling. Excessive speed increases the longitudinal strain at the edge. The reason is that the rapid travel of the side part increases the strain rate and produces work hardening, which has a significant impact on the stability of the product being formed. The friction coefficient is particularly important for controlling the surface quality, the actual production of lubricants, surface coating, and other means to reduce friction, to ensure the surface quality of the stainless steel. Too large a distance between stations affects the stability, because it reduces the binding force of the rolls. The analysis of the above data leads to the optimum parameters for the roll forming process to obtain a cross-sectional shape that meets the quality requirements.

## Figures and Tables

**Figure 1 materials-15-08023-f001:**
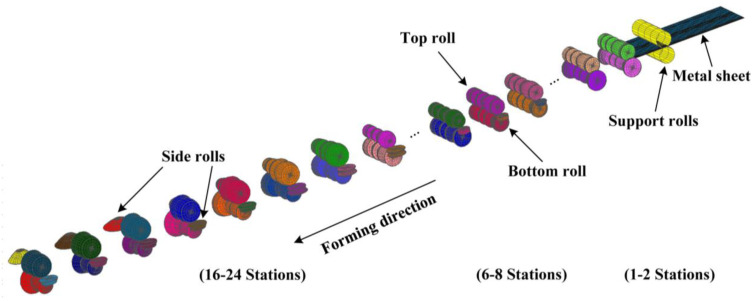
Assembly of cold roll forming.

**Figure 2 materials-15-08023-f002:**
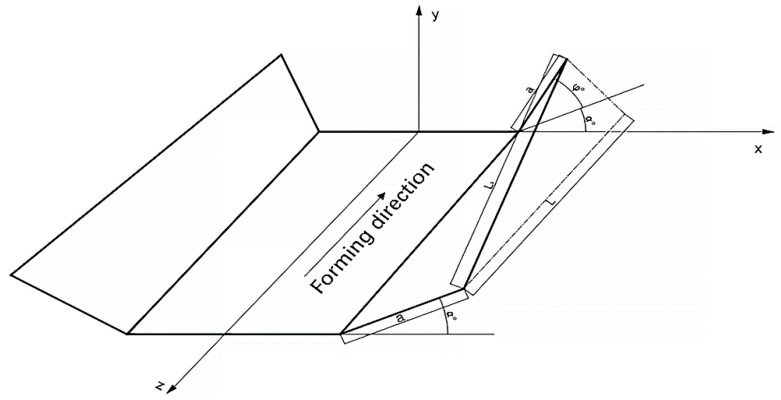
Flow of the metal strip.

**Figure 3 materials-15-08023-f003:**
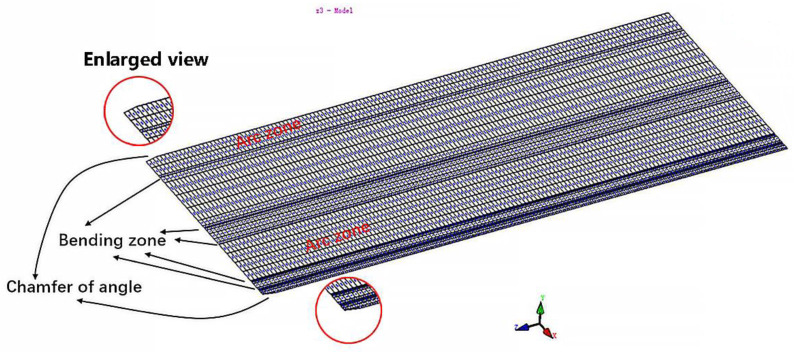
Strip meshing.

**Figure 4 materials-15-08023-f004:**
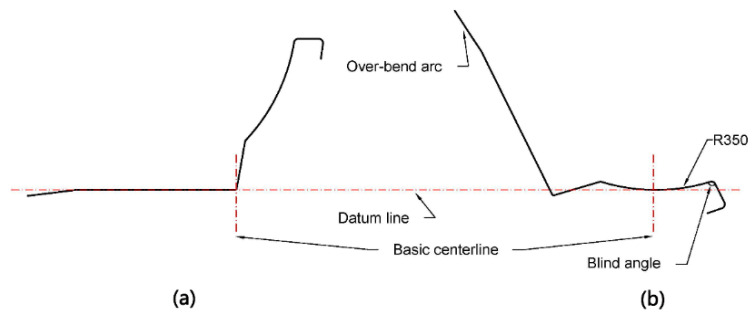
(**a**) Traditional method of placement; (**b**) optimal method of placement.

**Figure 5 materials-15-08023-f005:**
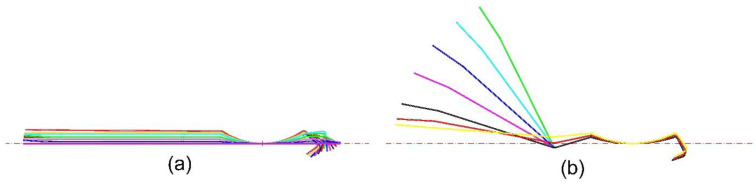
Flower pattern of (**a**) horizontal forming of 1–15 stations; (**b**) depth increasing in forming of 16–24 stations.

**Figure 6 materials-15-08023-f006:**
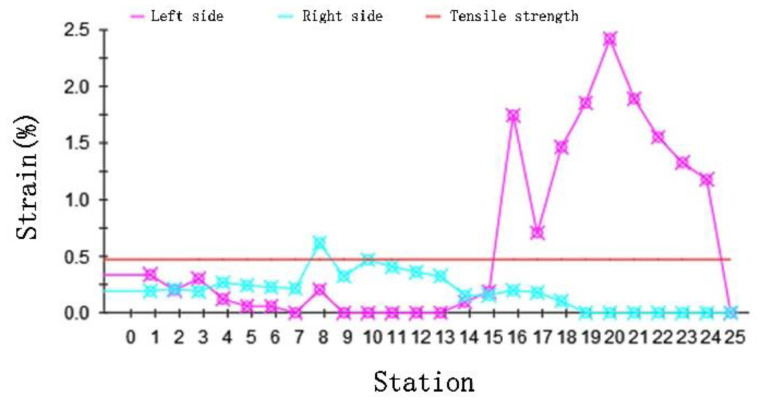
The curve of strain with station.

**Figure 7 materials-15-08023-f007:**
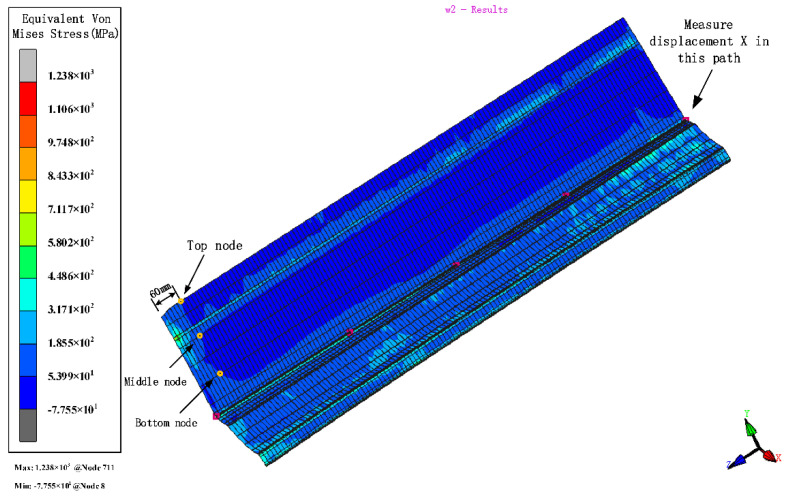
The variation in the equivalent von Mises stress distribution of strip.

**Figure 8 materials-15-08023-f008:**
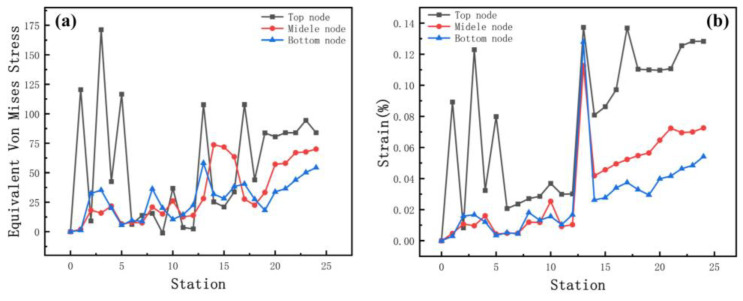
(**a**) Stress and (**b**) strain curves at different positions of the strip edge.

**Figure 9 materials-15-08023-f009:**
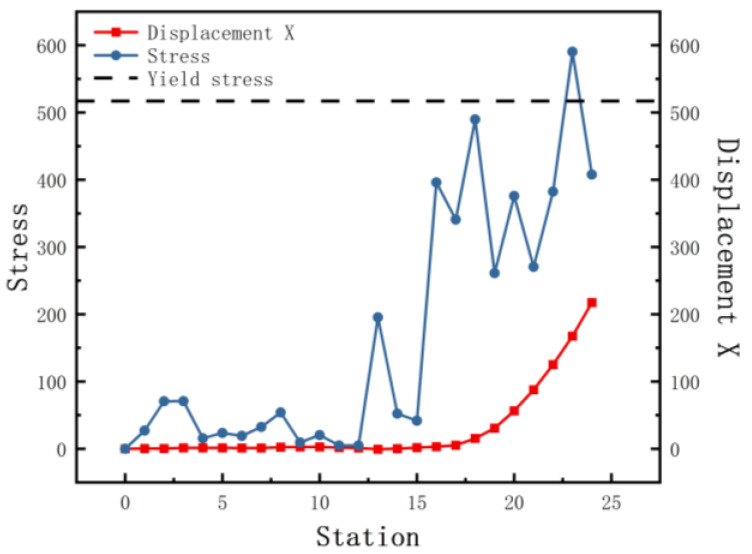
The equivalent von Mises stress and displacement X with stations.

**Figure 10 materials-15-08023-f010:**
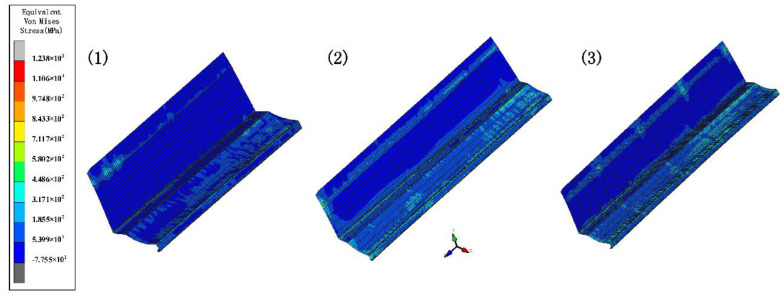
The variation in the equivalent von Mises stress distribution of strip under three cases.

**Figure 11 materials-15-08023-f011:**
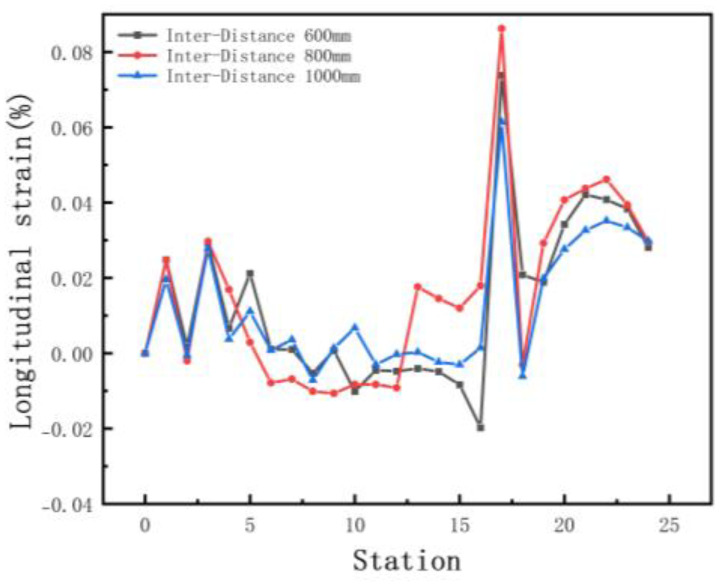
Longitudinal strain of strip in 3 inter−distance conditions.

**Figure 12 materials-15-08023-f012:**
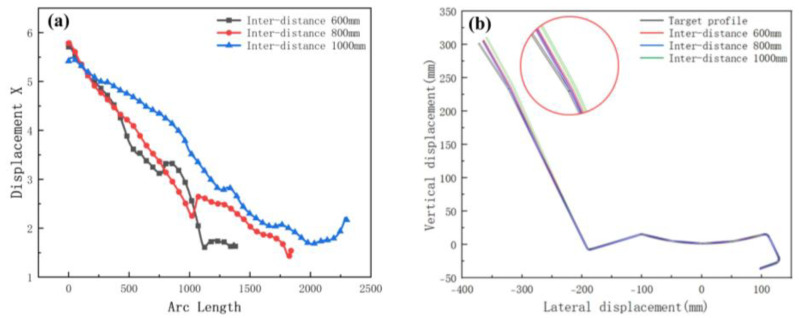
(**a**) Displacement X of strip in 3 inter-distance conditions; (**b**) cross-section of 3 inter-distance conditions.

**Figure 13 materials-15-08023-f013:**
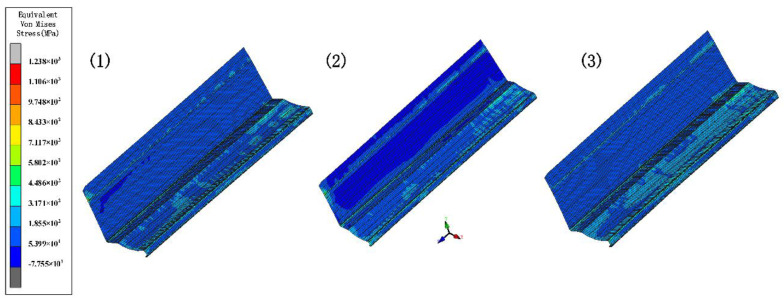
The variation in the equivalent von Mises stress distribution of strip under different friction levels.

**Figure 14 materials-15-08023-f014:**
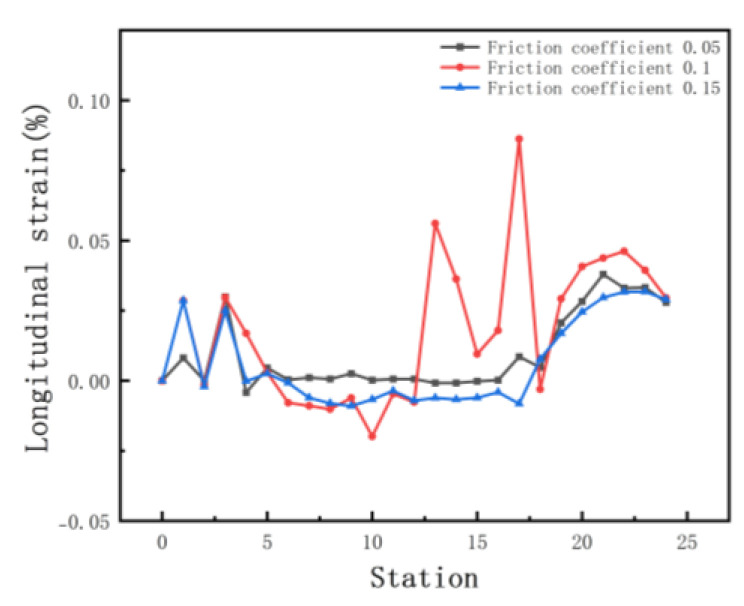
Longitudinal strain of strip in 3 friction coefficients.

**Figure 15 materials-15-08023-f015:**
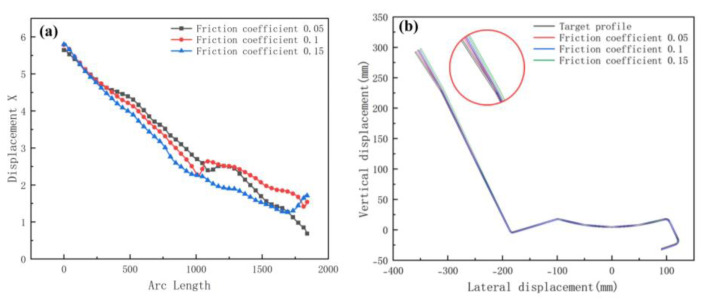
(**a**) Displacement X of strip in 3 friction coefficients; (**b**) cross-section of 3 friction coefficients.

**Figure 16 materials-15-08023-f016:**
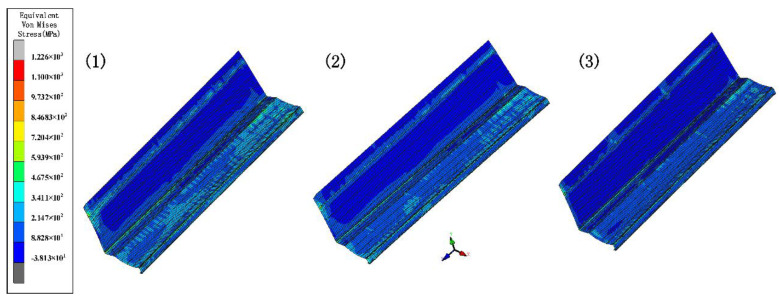
The variation in the equivalent von Mises stress distribution of strip under different diameter increments.

**Figure 17 materials-15-08023-f017:**
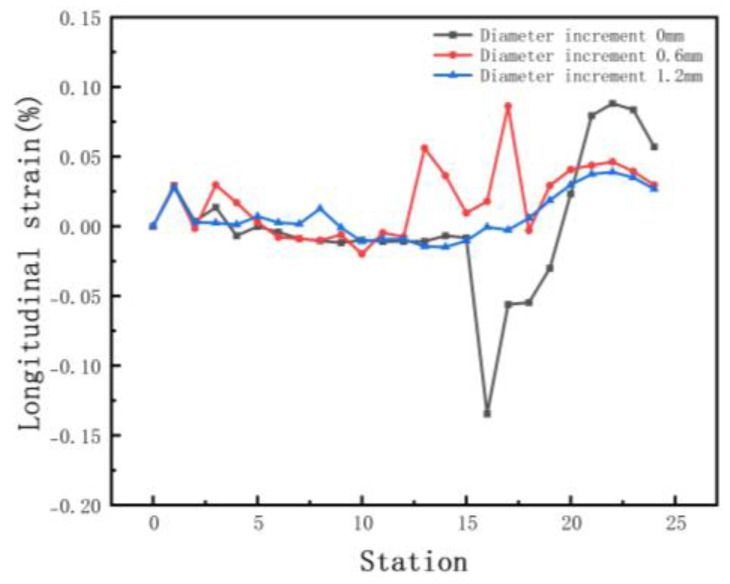
Longitudinal strain of strip in 3 diameter increments.

**Figure 18 materials-15-08023-f018:**
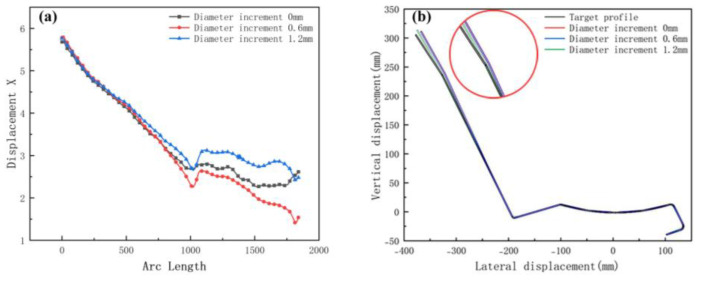
(**a**) Displacement X of strip in 3 diameter increments; (**b**) cross-section of 3 diameter increments.

**Figure 19 materials-15-08023-f019:**
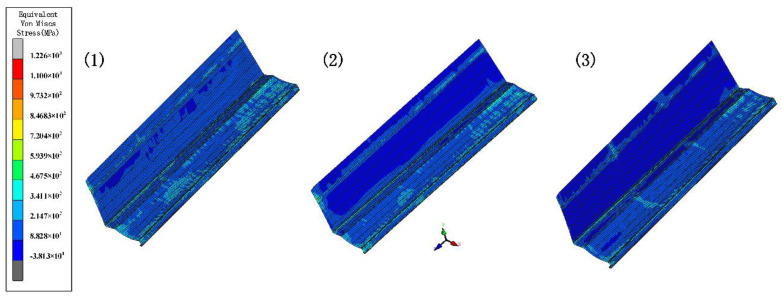
The variation in the equivalent von Mises stress distribution of strip under different velocities.

**Figure 20 materials-15-08023-f020:**
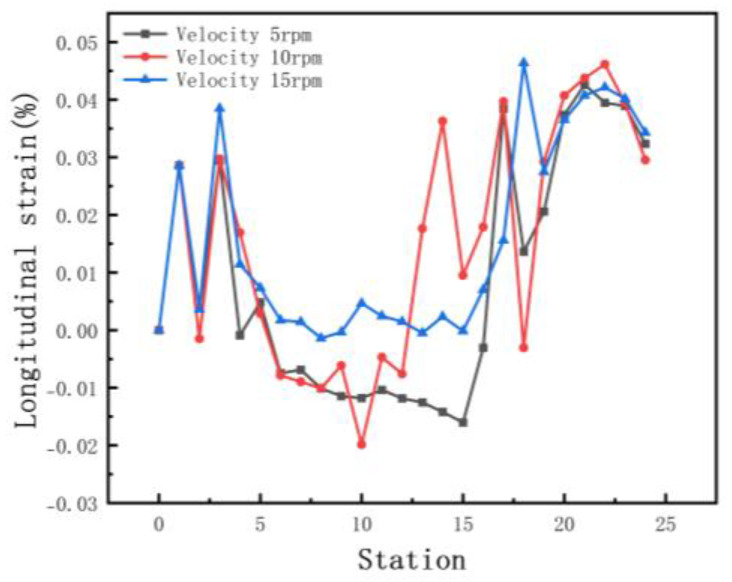
Longitudinal strain of strip in 3 velocities.

**Figure 21 materials-15-08023-f021:**
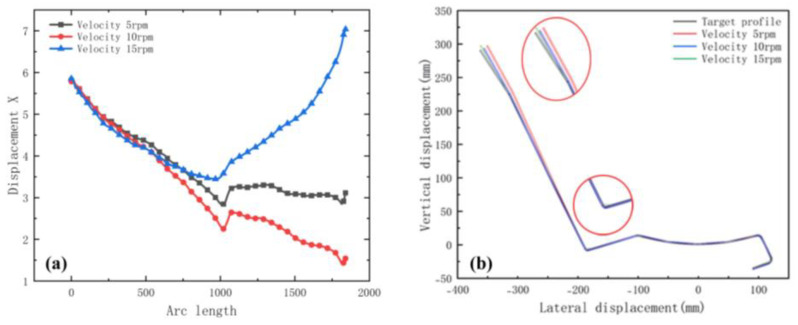
(**a**) Displacement X of strip in 3 velocities; (**b**) cross-section of 3 velocities.

**Figure 22 materials-15-08023-f022:**
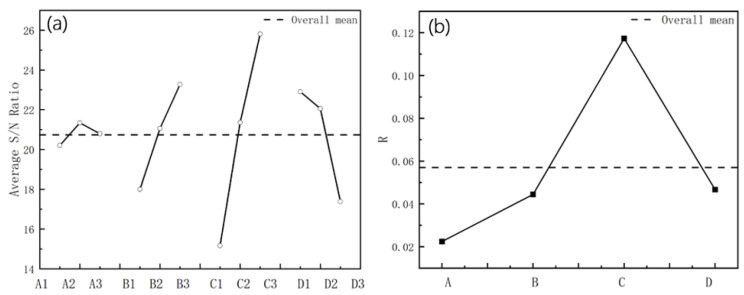
(**a**) The S/N ratio of 4 parameters; (**b**) The value of R with 4 parameters.

**Figure 23 materials-15-08023-f023:**
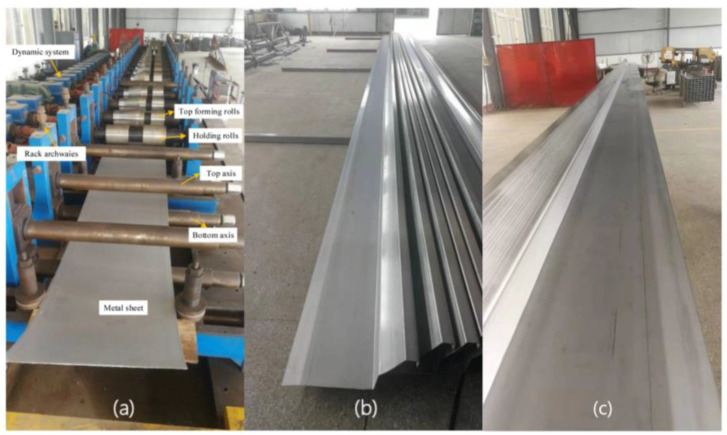
(**a**) The production line of roll forming process; (**b**,**c**) final products of metal strip.

**Table 1 materials-15-08023-t001:** Chemical composition of SUS301L-ST.

Element	C	Si	M	Cr	Ni	N	S	P
wt.%	≤0.03	≤1.00	≤2.00	16.00~18.00	6.00~8.00	≤0.20	≤0.03	≤0.045

**Table 2 materials-15-08023-t002:** Properties of SUS301L-ST.

Parameter	Unit	Value
Young’s modulus (E)	GPa	191.043
Poisson’s ratio (ϑ)	–	0.33
Yield stress ($\sigma_{s}$)	MPa	520.2
Ultimate tensile strength	MPa	881.027
Elongation	–	40%
Density	kg/m^3^	7850

**Table 3 materials-15-08023-t003:** Chemical composition of Cr12.

Element	C	Si	Mn	Cr	Ni	Co	Cu	S	P
wt.%	2.00~2.30	≤0.40	≤0.40	11.50~13.00	≤0.25	≤1.00	≤0.30	≤0.03	≤0.03

**Table 4 materials-15-08023-t004:** Different inter-distance cases.

Cases	1–15 Stands	16–24 Stands
1	600 mm	600 mm
2	600 mm	800 mm
3	600 mm	1000 mm

**Table 5 materials-15-08023-t005:** Parameters and their respective levels.

Parameter	Level 1	Level 2	Level 3
Inter-distance between roll stations (A)	600 mm	800 mm	1000 mm
Friction coefficient (B)	0.05	0.1	0.15
The diameter difference of rolls (C)	0 mm	0.6 mm	1.2 mm
Line velocity (D)	5 rpm	10 rpm	15 rpm

**Table 6 materials-15-08023-t006:** Standard orthogonal array for the experiments and results.

Experiments	Parameters				Maximum Strain (%)	S/N
	A	B	C	D		
1	600	0.05	0	5	0.20073	13.94775
2	600	0.1	0.6	15	0.13084	17.66519
3	600	0.15	1.2	10	0.03544	29.01013
4	800	0.15	0	15	0.18147	14.82390
5	800	0.05	0.6	10	0.09542	20.40721
6	800	0.1	1.2	5	0.03639	28.78039
7	1000	0.1	0	10	0.14553	16.74095
8	1000	0.15	0.6	5	0.05017	25.99112
9	1000	0.05	1.2	15	0.10410	19.65099

**Table 7 materials-15-08023-t007:** The result of range.

Results	Parameters			
	A	B	C	D
K_1_	0.36701	0.40025	0.52773	0.28729
K_2_	0.31328	0.31276	0.27643	0.27639
K_3_	0.29980	0.26708	0.17593	0.41641
k_1_	0.12234	0.13342	0.17591	0.09576
k_2_	0.10443	0.10425	0.09214	0.09213
k_3_	0.09993	0.08903	0.05864	0.1388
R	0.02241	0.04439	0.11727	0.04667

## Data Availability

The datasets used or analyzed during the current study are available from the corresponding author on reasonable request.

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
