# Peer review of "Cold Roll Forming Process Design for Complex Stainless-Steel Section Based on COPRA and Orthogonal Experiment"

_materials, 2022, doi:10.3390/ma15228023_

Round 1

Reviewer 1 Report

The article is interesting but it needs the following improvements:

·         the introduction does not allow the assessment of the current knowledge in the discussed field because the cited literature covers the last 5 years only in 30%, the remaining ones are publications from before 2017. The introduction should be supplemented with the latest research results

·         the purpose of the research should be clearly defined; what makes your research stand out from other works in this field

·         some figures are illegible, especially figs. 10, 13, 16, 19, in which color scale was inadequately selected; there should be [MPa] not [Mpa].

·         detailed comments have been made directly in the PDF file

Author Response

Manuscript ID: materials-1999804
Type of manuscript: Article
Title: Cold roll forming process design for complex stainless-steel section
based on COPRA and orthogonal experiment
Authors: Jing Wang, Hua-Min Liu *, She-Fei Li, Wan-Jun Chen

Dear Reviewer 1,

Thank you very much for your comments and suggestions. Those comments are all valuable and helpful for improving our paper. We have studied comments carefully and have corrected some unclear description. We have tried our best to improve and made some changes in the manuscript. Responds to the Reviewer 1 comments,

  1. the introduction does not allow the assessment of the current knowledge in the discussed field because the cited literature covers the last 5 years only in 30%, the remaining ones are publications from before 2017. The introduction should be supplemented with the latest research results

Respond: The introduction of our original paper is not innovative and didn’t emphasize the propose. We have added some new literatures to strengthen the innovation. Thank you very much for your precise advice. (page 2, line 47-50, page 2, line 81-84)

  1. the purpose of the research should be clearly defined; what makes your research stand out from other works in this field

Respond: We added something about the purpose of the research which made the purpose more complete. (page 2, line 85-87)

  1. some figures are illegible, especially figs. 10, 13, 16, 19, in which color scale was inadequately selected; there should be [MPa] not [Mpa].

Respond: We’re very sorry for our careless about the figures. We have corrected figures that you mentioned. Thank you for your carefully revising.

  1. detailed comments have been made directly in the PDF file

Respond: Thanks again for your careful review. We have corrected all the details which you mentioned in the PDF.

We tried our best to improve and revise the manuscript. Special thanks to you for your time and good comments. We hope that the revised manuscript is accepted for publication.

Your sincerely,

Authors

Reviewer 2 Report

1. Chemical composition of SUS301L-ST steel is very desirable (page 3, line 115).

2. Figure 3: The mesh was not the same for the whole strip? Or it is the same but looks different due to chamfering?

3. Chemical composition of Cr12 steel is also desirable (page 5, line 162).

4. Authors mention that results are consistent with theory (page 6, lines 186-187). Is it possible to describe this consistency briefly or provide some reference describing this theoretical concept?

5.  Figures 7,10,13,16,19 show equivalent stress distribution. Please, provide formula for calculation of this parameter.

6. Is it possible to give horizontal line representing yield strength value (page 7, line 210) on Figure 9?

7. Page 8, line 232. You write that strain becomes bigger. Please, specify it in number. Page 8, line 244. You write that displacement is larger. Please, characterize it numerically.

8. Page 12, line 320. You write that difference in equivalent stress is small. How much is it exactly?

9. Page 14, lines 358-362. Authors mention positive and negative correlation. If correlation coefficients were calculated, please, provide their values.

10. Page 14, lines 378-379. You mention compliance to precision and surface quality. Is it possible to describe it numerically?

11. Please, provide information concerning the equipment presented on Figure 23 (location, type of the equipment and other).

12. To emphasize obtained results, some numerical data concerning FEM simulation results and industrial trial could be added in the Conclusion section.

Author Response

Manuscript ID: materials-1999804
Type of manuscript: Article
Title: Cold roll forming process design for complex stainless-steel section
based on COPRA and orthogonal experiment
Authors: Jing Wang, Hua-Min Liu *, She-Fei Li, Wan-Jun Chen

Dear Reviewer 2,

Thank you very much for your comments and suggestions. Those comments are all valuable and helpful for improving our paper. We have studied comments carefully and have corrected some unclear description. We have tried our best to improve and made some changes in the manuscript. Responses to the Reviewer 2 comments, 

  1. Chemical composition of SUS301L-ST steel is very desirable (page 3, line 115).

Response: As reviewer suggested that the Chemical composition of SUS301L-ST is attached (page 3-4, line 122-123)

  1. Figure 3: The mesh was not the same for the whole strip? Or it is the same but looks different due to chamfering?

Response: The mesh deformed because of chamfering. We added enlarged views of chamfers to make the meshing clear. (page 4, line 132)

  1. Chemical composition of Cr12 steel is also desirable (page 5, line 162).

Response: As reviewer suggested that the Chemical composition of Cr12 is attached (page 5, line 171)

  1. Authors mention that results are consistent with theory (page 6, lines 186-187). Is it possible to describe this consistency briefly or provide some reference describing this theoretical concept?

Response: We’re very sorry for that is unclear here. We have corrected this part. (page 6, line 194-197)

  1. Figures 7,10,13,16,19 show equivalent stress distribution. Please, provide formula for calculation of this parameter.

Response: Because the equivalent stress distribution is obtained directly through the software, we have added the basic formula for Von mises yield criteria and its simple explanation here. (page6, line201-206)

  1. Is it possible to give horizontal line representing yield strength value (page 7, line 210) on Figure 9?

Response: We added the horizontal line representing yield strength value. This measuring position is a representative of the whole metal sheet. The stress starts in this position and the metal strip is gradually bent and deformed through the accumulated stresses. (page7, line226-229)

  1. Page 8, line 232. You write that strain becomes bigger. Please, specify it in number. Page 8, line 244. You write that displacement is larger. Please, characterize it numerically.

Response: We're sorry for the trouble caused by the lack of instructions and unclear description. we reconstituted the language in page8 line250-252. And we added a description of the metal strip. (page9, line263-265)

  1. Page 12, line 320. You write that difference in equivalent stress is small. How much is it exactly?

Response: We added some instructions using Specific values in this position. (page13, line339-342)

  1. Page 14, lines 358-362. Authors mention positive and negative correlation. If correlation coefficients were calculated, please, provide their values.

Response: We didn’t calculate the coefficients before, and we have calculated and added the coefficients in page15, line 383-385. The addition of the coefficients makes the results more convincing, thank you for your helpful advice.

  1. Page 14, lines 378-379. You mention compliance to precision and surface quality. Is it possible to describe it numerically?

Response: We are very sorry that we can't describe the surface quality numerically, because the surface quality in production is for the inspector to see whether there are indentation and scratches. And we added an additional figure for surface. Figure 23(c).

  1. Please, provide information concerning the equipment presented on Figure 23 (location, type of the equipment and other).

Response: We added some illustration in the figure of equipment. Figure 23(a).

  1. To emphasize obtained results, some numerical data concerning FEM simulation results and industrial trial could be added in the Conclusion section.

Response: we agree with your comments, which can make the simulation even more convincing. And we are very willing to supplement this experiment. But since qualified products have been assembled on railway, this can prove that the product accuracy and surface quality meet the requirement. And we added a figure of forming product which already in using. Figure 23(c).

We tried our best to improve and revise the manuscript. Special thanks to you for your time and good comments. We hope that the revised manuscript is accepted for publication.

Your sincerely,

Authors

Round 2

Reviewer 2 Report

In my view, paper can be recommended for publishing.